# Spectrophotometric-Based Assay to Quantify Relative Enzyme-Mediated Degradation of Commercially Available Bioplastics

**DOI:** 10.3390/polym15112439

**Published:** 2023-05-24

**Authors:** Matthew Hoekstra, Myron L. Smith

**Affiliations:** 1Institute of Biochemistry, Faculty of Science, Main Campus, Carleton University, Ottawa, ON K1S 5B6, Canada; myronsmith@cunet.carleton.ca; 2Department of Biology, Faculty of Science, Main Campus, Carleton University, Ottawa, ON K1S 5B6, Canada

**Keywords:** assay development, biopolymer degradation, bioplastic, enzyme degradation, proteinase K, PLA depolymerase

## Abstract

We present a spectrophotometric-based assay to identify enzymes that degrade commercially available bioplastics. Bioplastics comprise aliphatic polyesters with hydrolysis-susceptible ester bonds and are proposed as a replacement for petroleum-based plastics that accumulate in the environment. Unfortunately, many bioplastics can also persist in environments including seawater and waste centers. Our assay involves an overnight incubation of candidate enzyme(s) with plastic, followed by A610 spectrophotometry using 96-well plates to quantify both a reduction in residual plastic and the liberation of degradation by-products. We use the assay to show that Proteinase K and PLA depolymerase, two enzymes that were previously shown to degrade pure polylactic acid plastic, promote a 20–30% breakdown of commercial bioplastic during overnight incubation. We validate our assay and confirm the degradation potential of these enzymes with commercial bioplastic using established mass-loss and scanning electron microscopy methods. We show how the assay can be used to optimize parameters (temperature, co-factors, etc.) to enhance the enzyme-mediated degradation of bioplastics. The assay endpoint products can be coupled with nuclear magnetic resonance (NMR) or other analytical methods to infer the mode of enzymatic activity. Overall, the screening capacity of the spectrophotometric-based assay was demonstrated to be an accurate method to identify bioplastic-degrading enzymes.

## 1. Introduction

Due to environmental persistence and carbon emissions associated with petroleum-based single-use plastics, biodegradable bioplastics are being integrated into a variety of sectors and countries [1,2,3]. Biodegradable bioplastics are commonly created with plant-based materials that readily decompose much as plant materials are degraded in the natural world [2,4]. Industries such as medicine, agriculture, packaging, manufacturing and waste management have adopted the use of biodegradable bioplastic polymers for more eco-friendly applications compared to their petroleum-based plastic counterparts [1,2]. Unfortunately, bioplastic polymers/compostable bioplastic materials are not readily composted under normal waste-handling systems and are seen as contamination by many composting systems. For example, the common ‘compostable’ plastics composed of polylactic acid (PLA) and polyhydroxyalkanoate (PHA) are estimated to require 1–20 or sometimes 100+ years to decompose in open environmental settings [2,4]. PLA, PHA and other bioplastics can require specific environmental conditions to optimize environment-mediated degradation. For example, PLA is shown to degrade optimally in moist environments of 60 °C or higher [4]. Although most compost piles reach optimal temperatures in the thermophilic range of 45–65 °C, in northern climates such as Canada and the northern US, even established compost piles do not reach the necessary temperatures during fall, winter and spring months [5,6]. As a result, there is an apparent need to improve either the long-term or initial degradation of bioplastics to aid in decomposition in a compost or waste environment.

Generally, most biodegradable bioplastic polymers consist of a variety of aliphatic polyesters wherein ester bonds are readily hydrolyzable leading to long-term degradation [2,7]. The common biodegradable bioplastic polymers include but are not limited to PLA, PHA, poly-caprolactone (PCL), poly-propylene carbonate (PPC), poly-butylene succinate (PBS) and poly-butylene succinate-*co*-adipate (PBSA). These are all aliphatic polyesters, but their individual chemical structure and composition, as well as their hydrophobicity, can affect degradation processes [2,7]. Along with chemical properties, physical properties such as crystal structure, melting temperature (T_m_) and glass transition temperature (T_g_) can also affect how these bioplastic polymers degrade [2].

The aliphatic polyester nature of a majority of biodegradable bioplastic polymers means they are susceptible to enzymatic hydrolysis [2,7]. As such, recent research has focused on families of hydrolases and how they can facilitate enzymatic hydrolysis leading to bioplastic degradation [7]. Enzymes shown to have off-target activity towards pure bioplastics include cutinase-like enzymes, Proteinase K, Proteases, Esterases and Lipases [7,8,9,10,11,12,13,14]. Esterases, lipases and cutinase-like enzymes are all families of the hydrolase class of enzymes and therefore share similar catalytic mechanisms for their desired substrates. These hydrolytic enzymes can target proteins, lipids and carbohydrate biomolecules and polymers for degradation [15]. This is facilitated by using water to reduce a specific chemical bond (e.g., ester group), liberating a functional group or repeating unit [15]. Different families of hydrolases will target different chemical bonds (e.g., esterases vs. lipases). Proteases can effectively degrade proteins via the hydrolysis of peptide bonds, and most have been characterized to target the α-peptide bond between amino acids [16]. Although some proteases have been characterized to be very specific towards specific amino acids and or/ amino acid-binding epitopes, most proteases such as proteinase K have been demonstrated to be relatively non-specific for other substrates [16]. As a result, some proteases can catalyze the hydrolysis of chemical bonds in substrates in addition to proteins. 

Generally, candidate enzymes have been either isolated from bacteria associated with bioplastics in landfills or are hydrolases shown to have ‘off-target’ activity towards bioplastics because of chemical structure-based homology with their natural substrates [7]. For example, PLA depolymerase is an enzyme isolated from soil samples in Japan that was shown to degrade bioplastics such as PLA and PBSA [17,18]. Factors such as temperature, time, pH, co-factors and agitation may decrease, maintain or increase the ability of an enzyme to degrade bioplastics. For example, proteinase K has optimal enzymatic activity at 37 °C but can maintain upwards of 80% of activity from 20 °C to 60 °C and remains stable beyond 60 °C [19]. Moreover, Ca^2+^ concentration has also been shown to affect proteinase K stability and activity [20,21]. Examples such as these demonstrate the need to investigate whether specific reaction conditions could facilitate an enzyme’s overall ability to target bioplastics in an in vitro setting.

There currently exists a handful of different methods to assess enzymatic–plastic degradation. One commonly employed method is through mass loss measurements [12,14,22]. Simply, one can measure the mass of starting bioplastic materials against the remaining mass after enzyme incubation. A dose–responsive relationship is a further indication of enzyme activity [12,14,22]. Spectrophotometric-based assays have been developed but only with aromatic ring-containing plastics [23]. For example, it has been shown that the enzyme-mediated degradation of polyethylene terephthalate (PET) can be quantified via bulk A_260_ readings, as absorbance measurements of 260 nm light have been used for the detection of PET by-products, TPA, MHET and BHET [23]. Other methods such as bioplastic–agar emulsification have been developed to screen bacteria for their bioplastic-degrading capabilities by measuring the turbidity of bioplastic–agar at specific wavelengths [8,9,10,17,24]. Although this method is ideal for screening bacteria, once a candidate enzyme is discovered, other techniques need to be implemented to analyze the enzymatic degradation of bioplastics [8]. These techniques, such as scanning electron microscopy (SEM), mass spectrometry analysis and nuclear magnetic resonance (NMR) spectroscopy, have been used to validate an enzyme’s ability to degrade pure bioplastic species [8,9,12,14,22]. These analytical techniques require specialized equipment and therefore are not ideal for routine enzyme screening purposes. Other indirect methods to evaluate the enzymatic degradation of bioplastics include gel permeation chromatography analysis [8,9,22], differential scanning calorimetry and the measuring CO_2_ production as it is a by-product of common bioplastic degradation processes [12,13,22]. These indirect methods can be labor- and material-intensive, require specialized equipment and be cost-prohibitive and therefore are not ideal for screening for enzymes that degrade bioplastics.

The discovery of enzymes that target bioplastics in an in vitro setting is critical in helping to identify other potential ways to aid in the degradation of bioplastics, either in pre-landfill or landfill settings. However, the majority of the studies have been conducted on pure species of bioplastics, and the common commercially available biodegradable bioplastics used in everyday household and industrial settings are more often mixtures of various species of plastics [8,9,12,17,18]. As such, investigations into candidate enzymes that degrade commercially available biodegradable bioplastic mixtures are required, in the same or similar sense that they have been demonstrated to target and degrade pure bioplastic samples in vitro [7,8,9,10,11,12,13,14,17,18]. Moreover, as traditional methods of evaluating the enzyme-mediated degradation of bioplastics are labor- and material-intensive, require specialized equipment and/or are cost-prohibitive, they are not suitable for the rapid screening of candidate or engineered enzymes. As such, a medium-to-high throughput method for screening an enzyme’s capability to degrade mixtures of bioplastics is required.

Here, we provide a semi-quantitative spectrophotometric approach to screen enzymes in a 96-well plate format that is applicable to real-world, complex bioplastic mixtures [8,9,12,17,18]. For this, we adopted standard methods using emulsifying solutions and surfactants within a low-volume format that can be semi-automated [8,9,10,17]. As long as the target bioplastic mixture can be dissolved in an organic solvent, one can use this assay to optimize enzyme and substrate concentrations and other reaction parameters. To demonstrate, we utilized the proteinase K enzyme and the PLA depolymerase enzyme in our experiments as positive controls due to their reported activity towards pure bioplastics such as PLA [17,18,25]. We were able to demonstrate that proteinase K and PLA depolymerase can both degrade commercially sourced bioplastic mixtures in comparison to no-enzyme controls as well as a negative control cell wall lysing enzyme mixture. We validated our spectrophotometric assay results with mass loss measurements and analytical techniques based on SEM and NMR. The spectrophotometric-based assay can be applied to accurately screen and characterize the enzymatic degradation of complex bioplastics.

## 2. Materials and Methods

### 2.1. Preparation of Biodegradable Plastic

Proteinase K and PLA depolymerase were previously shown to degrade pure poly-lactic acid (PLA) [8,13,17,18]. We developed an assay to test the efficacy of these enzymes to degrade the bioplastic within a commercially available certified compostable bag (Appendix A). For standard sample preparation, a single-hole-punch device was used to obtain 6 mm diameter discs (Appendix A). Generally, 1 disc was equal to 0.55 mg, and 2 discs (1.1 mg) were used per tube in spectrophotometric-based assays. All sample preparations were performed at room temperature. It should be noted that areas of the bags with printing were avoided.

### 2.2. Purification of PLA Depolymerase

A pET25b(+) plasmid containing a PLA depolymerase (obtained from Toshiaki Nakajima-Kambe, University of Tsukuba, Tsukuba, Japan [17,18]) was transformed in *Escherichia coli* strain BL21 (DE3)C+ for protein expression and grown in 400 mL LB medium. Cultures for protein expression were inoculated with 0.5–1% starter culture and grown with shaking (200 RPM) at 37 °C to an A_600_ of 0.5–0.6, whereupon the temperature was decreased to 16 °C and the expression was induced with the addition of 0.1 mM isopropyl ß-D-1-thiogalacttopyranoside (IPTG). Following IPTG induction, cultures were incubated for 5 h at 16 °C, and then cells were pelleted by centrifugation (5000× *g* for 15 min at 4 °C) and the medium was decanted. Cell pellets were washed once with ice-cold 1× PBS and pelleted as above by centrifugation. PBS was decanted and pellets were snap-frozen in liquid nitrogen and then stored at −80 °C.

For protein purification, pellets were resuspended in 20 mM NaHPO_4_, 10% glycerol, 150 mM NaCl and 0.5 mM DTT and supplemented with protease and phosphatase inhibitors (1 mM PMSF, 0.5 μM Aprotinin, 10 μM E-64, 1 μM Pepstatin). Cells were lysed via sonication 5 × 30 s on/off at 70% amplitude intensity on ice for a total of 45 min. The cell lysate was cleared via centrifugation (18,000× *g* for 45 min at 4 °C) and the soluble lysate was loaded onto a 600 μL Ni-NTA agarose resin (Qiagen, Cat#30210) affinity column at 4 °C. The affinity column was washed and eluted with an imidazole gradient (40–250 mM imidazole in 20 mM NaHPO_4_) and protein fractions were collected. Protein fractions were dialyzed for 3 h at 4 °C in a protein storage buffer (20 mM NaHPO_4_, 10% glycerol, 150 mM NaCl, 0.5 mM DTT) using 6–8 kDa molecular weight cut-off dialysis tubing (FisherScientific, Cat#08-670A). Protein purity was assessed through 12% SDS-PAGE followed by staining with Coomassie Brilliant Blue G-250 stain (Appendix A). Protein concentration was quantified through a standard Bradford assay [26]. Proteins were snap-frozen in liquid nitrogen and stored at −80 °C for later use.

### 2.3. Spectrophotometric-Based Screening Assay

All proteinase K (ThermoFisher, Cat#25530031) reactions took place in Tris-based buffer (50 mM Tris (pH 8.0), 0.1% Triton X-100). All PLA depolymerase reactions took place in protein storage buffer plus Triton X-100 (20 mM NaHPO_4_, 10% glycerol, 150 mM NaCl, 0.5 mM DTT, 0.1% Triton X-100). A mixture of lysing enzymes from *Trichoderma harzianum* (including β-glucanase, cellulase, protease and chitinases) (Sigma-Aldrich, Cat#L1412) served as a negative control for these experiments, and reactions took place in the same Tris-based buffer as proteinase K. To begin, 1.1 mg of bioplastic (2 discs) was added to 1.5 mL Eppendorf tubes with each respective reaction buffer and enzyme. No-enzyme controls were also set up for each enzyme-specific buffer. Reactions proceeded for 24 h at 37 °C in a tube revolver rotator (ThermoFisher, Cat#88881001). After overnight incubation, the supernatant from each reaction was transferred to a new 1.5 mL tube. In total, 1 mL of Tris-based buffer was added to the residual plastic from each reaction. The separate tubes containing residual plastic and decanted supernatant each had 100 μL of CHCl_3_ added, followed by vortexing until each solution turned from transparent to turbid. Once turbid, all samples were sonicated using a VCX 130 ultrasonic processor (Sonics & Materials INC, Newtown, CT) for 1 min at 65% amplitude until organic and aqueous phases were emulsified. Once emulsified, 200 μL from each reaction tube was added to a 96-well clear microplate (Corning, Cat#353072) in n = 4 replicates. The turbidity of the solution was measured at an absorbance of 610 nm and was read immediately using a Cytation 5 plate reader (Bio-Tek, Winooski, VT, USA).

For proteinase K, a 4× dilution curve from 4000–0 μg/mL was used to determine a dose–response regression and an optimal concentration for subsequent experiments. Two biological replicates each with four technical replicates were used. For PLA depolymerase, a 10× dilution curve from 1000–0 μg/mL was used to determine a dose–response regression and an optimal concentration moving forward. Two biological replicates each with four technical replicates were used. With the lysing enzyme mixture from *T. harzianum,* a 4× dilution curve from 4000–0 μg/mL was used to validate the efficacy of both assays as a negative control. Two biological replicates each with four technical replicates were used.

For temporal studies assessing proteinase K and PLA depolymerase activity towards bioplastic substrates, the assays were set up as described. Briefly, reactions were initiated at 6 h intervals. Proteinase K positive control samples were at a concentration of 4000 μg/mL in the Tris-based buffer (50 mM Tris (pH 8.0), 0.1% Triton X-100). The no-enzyme control (0 μg/mL proteinase K) samples were incubated in only Tris-based buffer. For each condition, two biological replicates each with four technical replicates were used. PLA depolymerase positive control samples were at a concentration of 1000 μg/mL in protein storage buffer plus Triton X-100 (20 mM NaHPO_4_, 10% glycerol, 150 mM NaCl, 0.5 mM DTT, 0.1% Triton X-100). The no-enzyme control (0 μg/mL PLA depolymerase) samples were incubated in only protein storage buffer. After each incubation time, the rest of the assay was completed as described above. For each condition, two biological replicates each with four technical replicates were used.

For assessing proteinase K activity at different temperatures, the assay was set up the same as described above, except tubes were in incubators set at 37 °C and 65 °C for 24 h. For assessing the Ca^2+^ effects on the proteinase K activity, reactions were set up as described above for the enzyme-only control, and the other two reactions were supplemented with 1 mM CaCl_2_ or 1 mM CaCl_2_ + 2 mM EDTA from 1 M stock solutions, still totaling 1 mL in total volume.

### 2.4. Enzyme-Mediated Mass Loss

Mass loss measurements with commercially available biodegradable plastic were evaluated by incubating enzymes with the bioplastic for 24 h. To begin, approximately 5 mg of bioplastic (9 discs) was added to a 1.5 mL Eppendorf tube and initially dried overnight at ~50 °C. After overnight drying, the initial starting mass of tubes containing bioplastic was weighed using an analytical balance (Sartorius Canada Inc, Oakville, ON). Proteinase K, PLA depolymerase or lysing enzymes from *T. harzianum* (including β-glucanase, cellulase, protease, and chitinases as a negative control) were separately added to tubes containing weighed bioplastic discs. The same associated buffers used in the spectrophotometric screening assays were used in mass loss assessments to maintain reaction consistency. No-enzyme controls were also set up using each enzyme-respective buffer. Reactions were incubated for 24 h on a tube revolver rotator at 37 °C. After overnight incubations, the supernatants were removed from the tubes and the remaining plastic was gently washed 3× with 1 mL of deionized water. After the last wash, the remaining bioplastic was dried overnight at ~50 °C, and the final remaining mass of tubes containing bioplastic was weighed using the analytical balance. The difference was calculated between the starting and final mass of the tubes and plastic and presented.

For proteinase K, a 4× dilution curve from 4000 to 0 μg/mL was used to determine a dose–responsive regression of mass loss and an optimal concentration moving forward with other experiments. For PLA depolymerase, a 10× dilution curve from 1000 to 0 μg/mL was used to determine a dose–responsive regression of mass loss and an optimal concentration moving forward. For the lysing enzyme mixture from *T. harzianum,* a 4× dilution curve from 4000 to 0 μg/mL was used as a negative control. Three technical replicates were used for all enzymes.

For temporal bioplastic mass loss studies, proteinase K and PLA depolymerase assays were set up as described. Briefly, reactions were initiated at 6 h intervals. Proteinase K positive control samples were at a concentration of 4000 μg/mL in the Tris-based buffer (50 mM Tris (pH 8.0), 0.1% Triton X-100). The proteinase K no-enzyme control (0 μg/mL) samples were incubated in only Tris-based buffer. PLA depolymerase positive control samples were at a concentration of 1000 μg/mL in protein storage buffer plus Triton X-100 (20 mM NaHPO_4_, 10% glycerol, 150 mM NaCl, 0.5 mM DTT, 0.1% Triton X-100). The PLA depolymerase no-enzyme control (0 μg/mL) samples were incubated in only protein storage buffer. After overnight incubation, the supernatants were removed from the tubes and the mass loss was determined as described above. Three technical replicates were used for all enzymes.

### 2.5. Scanning Electron Microscopy (SEM)

All in vitro enzymatic assays to detect bioplastic degradation via SEM were performed as described. All proteinase K reactions took place in Tris-based buffer (50 mM Tris (pH 8.0), 0.1% Triton X-100). All PLA depolymerase reactions took place in protein storage buffer plus Triton X-100 (20 mM NaHPO_4_, 10% glycerol, 150 mM NaCl, 0.5 mM DTT, 0.1% Triton X-100). As was performed for the spectrophotometric-based assay, all SEM-based experiments were performed with 1.1 mg of bioplastic (2 discs). To begin, 1.1 mg of bioplastic (2 discs) was added to 1.5 mL Eppendorf tubes with each respective reaction buffer and enzyme. Reactions proceeded for 24 h at 37 °C. After overnight incubation, the supernatants were removed from the tubes and any remaining plastic was gently washed 3× with 1 mL of deionized water. After the last wash, the remaining bioplastic was dried overnight at ~50 °C and prepared for SEM in the Carleton University Nano Imaging Facility. Briefly, dried samples were coated with a gold layer (~10 nm) for conductive purposes using a Quorum Q150T ES sputter and then imaged using a scanning electron microscope (TESCAN VegaII XMU) at 1 kV acceleration voltage in high vacuum mode. The ImageJ “Analysis of particles” feature was used to quantify holes in the bioplastic indicative of enzyme degradation [27].

### 2.6. Nuclear Magnetic Resonance (NMR)

All in vitro enzymatic assays to detect bioplastic degradation via NMR were performed as described (Appendix A). All proteinase K reactions took place in Tris-based buffer (50 mM Tris (pH 8.0)). All PLA depolymerase reactions took place in protein storage buffer (20 mM NaHPO_4_, 10% glycerol, 150 mM NaCl, 0.5 mM DTT). So as not to interfere with NMR, Triton-X100 was omitted from the reaction buffers. Similar to the spectrophotometric-based assays, all NMR-based experiments were performed with 1.1 mg of bioplastic (2 discs). To begin, 1.1 mg of bioplastic (2 discs) was added to 1.5 mL Eppendorf tubes with each respective reaction buffer and enzyme. Reactions proceeded for 24 h at 37 °C. After overnight incubation, the supernatant from each reaction was removed using glass pipettes and put into a new 1.5 mL tube. The supernatant was passed through a 10 kDa spin column (MilliporeSigma, Cat#UFC801008) as recommended by the manufacturer to remove enzymes that may interfere with NMR signals. Samples were roto-evaporated for 2 h to remove the remaining aqueous solution and processed in the Carleton University NMR Facility. Briefly, all NMR spectra were obtained at 7.05 T (ν0 = 300.15 MHz) on a Bruker Avance 300 spectrometer. Briefly, 1H NMR spectra were obtained in CDCl_3_ using a zg30 pulse sequence with a relaxation delay of at least 1 s and an acquisition time of 3.5 s. In total, 64–256 transients were collected. Then, 1H NMR spectra were internally referenced to TMS.

### 2.7. Data Analysis

Paired two-tailed T-tests were used to examine the statistical significance of bioplastic degradation between enzyme and no-enzyme treatments in spectrophotometric-based assays and mass loss measurements. For the analysis of proteinase K activity in various reaction conditions with/without the presence of CaCl_2_/EDTA, a 2-way ANOVA with concentration and absorbance held as integers and reaction condition (presence/absence of CaCl_2_ and EDTA) held as a factor was used to determine statistical significance between reaction conditions. The quantification of “holes” in bioplastic via enzymatic-mediated degradation was derived from the ImageJ “Analysis of particles” feature [27]. ImageJ allows a user-defined threshold that can be set relative to the pixels attributed with a positive control, in this case, a “hole” produced via enzyme-mediated degradation. The picture was set to 8-bit and the threshold was set to >99.3% of the total signal on the side of the dark signal. The size was set from 0.0002 to 2.00 inch^2^ and the circularity was set from 0.1 to 1.00 to minimize false positives.

## 3. Results

To design an in vitro assay to screen the enzymatic-mediated degradation potential of bioplastics, we combined methods in which plastic was emulsified in a solid medium with growth nutrients and agar [8,9,10,17]. Briefly, by applying CHCl_3_ to a buffered solution containing a small amount of bioplastic and a small percent of a detergent, the emulsification of the solution can be achieved through sonication. The result of the sonication turns a transparent solution containing an organic layer (bioplastic in CHCl_3_) and an aqueous layer (buffer in water) into a turbid emulsified solution (Figure 1). As the amount of bioplastic in the emulsified solution increases, the turbidity of the solution also increases. By pipetting 200 μL aliquots into wells of a clear 96-well plate and reading an absorbance wavelength of 610 nm, one can observe a dose–responsive relationship as the mass of bioplastic increases in this 1.1 mL volume (Figure 2). A wavelength of 610 nm was chosen based on previous studies that established a range of 580–660 nm to measure the turbidity of the emulsified bioplastics [9,11,18]. Ultimately, by comparing to a proper control/blank, this spectrophotometric-based assay can become a relatively quantifiable assay by reading the A_610_ of a solution containing an amount of plastic in the linear portion of the curve in Figure 2. As a result, for all spectrophotometric-based assays used for screening enzymes, we progressed with using 1.1 mg of plastic (two discs) as this represents approximately 75% of the maximum weight for the linear portion of the curve. 

This assay is convenient for medium-throughput enzyme screening. As CHCl_3_ is denser than water, it naturally settles at the bottom of an Eppendorf tube where the remaining bioplastic also settles. This is advantageous as CHCl_3_ can be added after the incubation of enzyme/no-enzyme treatments with bioplastics as a “detection reagent”. It is critical to add CHCl_3_ after enzyme incubation as CHCl_3_ may otherwise contribute to protein precipitation which could result in an inactive enzyme [28]. Moreover, upon the successful enzyme-mediated degradation of bioplastics, the supernatant of the initial enzymatic reaction contains by-products of bioplastic degradation. As such, CHCl_3_ can be added to the supernatant. The result is an emulsified solution that has increased turbidity from by-products of bioplastic produced as a result of degradation.

Once we determined the optimal amount of bioplastic (1.1 mg) to assay an enzyme’s bioplastic-degrading capacity (Figure 2), we then assessed the validity of two enzymes that have been reported to degrade pure biopolymers, such as PLA. The enzymes tested were proteinase K and PLA depolymerase (Figure 3) [8,13,17,18]. We also implemented a mixture of lysing enzymes from *T. harzianum* including β-glucanase, cellulase, proteases, and chitinases to be our negative control. To validate our absorbance-based assay and to ensure we are observing true degradation, we also assessed the bioplastic-degrading activity of these enzymes via mass-loss measurements (Figure 3). The mass before and after the incubation of the enzyme was recorded and the total mass loss was calculated based on changing enzyme concentrations. This type of assessment has been reported to be effective by other groups looking to screen the pure bioplastic-degrading capabilities of other enzymes [12,22]. It should be noted that all values in Figure 3 are shown as relative changes in either A_610_ values or in bioplastic mass since we observed a small amount of change in A_610_ or mass loss in bioplastic in our no-enzyme controls throughout this study. Given most bioplastic species can degrade naturally in aqueous environments due to the hydrolysis of ester bonds, the small decreases observed are likely due to basal levels of bioplastic degradation through the in-solution reactions [22,29,30,31].

To maximize insights from our spectrophotometric-based assay, we looked to detect plastic degradation by both a reduction in residual plastic and an increase in degradation products in supernatant compared to the no-enzyme controls (Figure 3). Using our spectrophotometric-based assay, we observed a reduction in the residual (non-degraded) bioplastic after 24 h incubation that correlated to increasing the enzyme concentration with both proteinase K and PLA depolymerase (Figure 3A). When compared to the respective controls, there were statistically significant decreases in A_610_ from proteinase K in concentrations > 62.5 μg/mL and PLA depolymerase concentrations > 100 μg/mL (*p* < 0.05). Collectively, the data indicate that treatments with these enzymes can significantly degrade commercial bioplastic within 24 h. In contrast, no significant reduction in residual bioplastic was observed in our negative control containing a mixture of lysing enzymes from *T. harzianum*.

In analyzing the decanted supernatant upon reaction completion, we observed an increase in relative A_610_ absorbance that correlated with proteinase K and PLA depolymerase concentrations (Figure 3B). When compared to the respective no-enzyme controls, there were statistically significant increases in A_610_ from proteinase K in concentrations > 62.5 μg/mL and PLA depolymerase concentrations > 10 μg/mL (*p* < 0.05). These data suggest that as the concentration of proteinase K and PLA depolymerase increased, there was increasing bioplastic degradation leading to increasing amounts of by-products of degradation released into the solution. A slight positive trend was also observed in our negative control containing the lysing enzymes from *T. harzianum* but was not significant compared to the no-enzyme control (*p* > 0.05). Ultimately, both sets of data from residual bioplastic (Figure 3A) and degradation products in the supernatant (Figure 3B) indicate that proteinase K and PLA depolymerase are able to degrade our commercially available biodegradable plastic bags in solution.

In order to further validate the ability of proteinase K and PLA depolymerase to target bioplastics for degradation in-solution, as well as the efficacy of the spectrophotometric-based assay, we analyzed the change in the mass of bioplastics before and after incubation with these enzymes in comparison to the no-enzyme controls (Figure 3C). A similar trend was observed to that in Figure 3A, such that as concentrations of proteinase K and PLA depolymerase increase, we observe a greater loss in the mass of bioplastic (Figure 3C). When compared to their respective controls, there were statistically significant decreases in the percent mass loss from proteinase K in concentrations > 62.5 μg/mL and PLA depolymerase concentrations > 100 μg/mL (*p* < 0.05). Specifically, using a concentration of 4000 μg/mL of proteinase K, we observed an average of ~21% (±1%) mass loss of starting bioplastic material (*p* < 0.05). Using PLA depolymerase we observed an average of ~32% (±3%) mass loss of starting bioplastic material using 1000 μg/mL of PLA depolymerase (*p* < 0.05). A slight negative trend was also observed in our negative control containing lysing enzymes from *T. harzianum* but it was not significant compared to the no-enzyme control (*p* > 0.05). Some background degradation was expected since many bioplastic species are reported to degrade naturally in aqueous environments [22,29,30,31]. Overall, the mass loss data confirms that the spectrophotometric-based assay is a quantitative assay that can be used to screen for enzymes with bioplastic degradation capabilities in a multi-well, medium-throughput format. Going forward, concentrations of 4000 μg/mL of proteinase K and 1000 μg/mL of PLA depolymerase were deemed optimal for future experiments.

To further assess how incubation time affects the relative activity of proteinase K and PLA depolymerase towards commercially available bioplastic, we assessed both enzyme end-point activities over 6 h intervals (Figure 4). Specifically, we assessed the relative bioplastic degradation at 0, 6, 12, 18 and 24 h with concentrations of 4000 μg/mL proteinase K and 1000 μg/mL PLA depolymerase. We also included no-enzyme controls to determine what effects each of the respective buffers had on the spectrophotometric-based assay and whether these results were represented in the mass loss assessments and consistent between all assays.

The spectrophotometric-based assay demonstrated an increase in the degradation of commercially available bioplastics as the incubation time increased with both proteinase K and PLA depolymerase enzymes. Generally, we observed that as incubation time increased in 6 h intervals there was an associated decrease in A_610_ values from the residual bioplastic (Figure 4A). Specifically for proteinase K compared to the control (0 h reaction with 4000 μg/mL proteinase K), we began to observe a significant decrease (*p* < 0.05) in the relative A_610_ values from as early as an initial 6 h incubation. PLA depolymerase when compared to its control (0 h reaction with 1000 μg/mL PLA depolymerase) demonstrated a significant decrease (*p* < 0.05) in relative A_610_ values from the residual bioplastic as early as 12 h. There were also observed decreases in the relative A_610_ values in the no-enzyme controls as the incubation time increased. Both sets of no-enzyme control data showed significant decreases (*p* < 0.05) in the relative A_610_ values from the remaining residual bioplastic from 18 and 24 h when compared to the no-enzyme control 0 h reaction (Figure 4A). This was expected and is consistent with our previous observations of some loss in mass in our no-enzyme controls (Figure 3C).

In analyzing the decanted supernatant at each time interval, we observed an increase in relative A_610_ values as incubation times increased with proteinase K and PLA depolymerase enzymes indicative of the release of degradation products into the solution (Figure 4B). Specifically, the relative A_610_ values significantly increased (*p* < 0.05) for both proteinase K and PLA depolymerase when the reaction time exceeded 6 h, in comparison to the respective controls (0 h reaction with enzyme present). Similar to the residual bioplastic analysis (Figure 4A), both sets of no-enzyme control data showed significant increases (*p* < 0.05) in relative A_610_ values from the supernatant from the 18 and 24 h reaction when compared to the no-enzyme control 0 h reaction (Figure 4B). This further supports that these bioplastics when in an aqueous environment can display some basal levels of degradation most likely attributed to the hydrolysis of ester bonds [22,29,30,31].

Similar to what was performed for our initial dilution curves (Figure 3C), we replicated our temporal experiments above using the mass-loss assessment (Figure 4C). For both proteinase K and PLA depolymerase when compared to their respective controls (0 h reaction with enzyme present), we began to observe a significant decrease (*p* < 0.05) in the relative percent mass from the 6 h incubation period to the complete 24 h reaction. Generally, we observed an approximately 4–5% decrease in the mass of the bioplastic per 6 h incubation period from the 6 h to 24 h incubation period. Here, we observed that over the initial 24 h incubation period, there is a temporal-based factor in enzyme-mediated bioplastic degradation which increases from 0 to 24 h of the allocated reaction time. Consistent with other data in this study, we observed a <6.5% decrease in mass loss in the no-enzyme control samples after 24 h. In both proteinase K and PLA depolymerase no-enzyme controls, only the 24 h incubation time displayed a significant loss (*p* < 0.05) in the remaining mass of the bioplastic compared to the no-enzyme 0 h incubation control. Again, this further indicates the basal levels of bioplastic degradation observed in our samples. Overall, we have demonstrated the utility of our spectrophotometric-based assay to temporally screen enzyme degradation processes.

To further validate the efficacy of our A_610_-based enzyme screening assay for enzyme-mediated bioplastic degradation, we sought to demonstrate through other analytical methods that we were in fact observing plastic degradation via proteinase K and PLA depolymerase. To help confirm enzyme-mediated plastic degradation, scanning electron microscopy was used to directly observe signs of bioplastic degradation. High- and medium-enzyme concentrations along the dilution curves used in both the spectrophotometric-based assay and the mass loss assessments (Figure 3) were chosen, along with a no-enzyme control. We observed a dose–responsive relationship between the increasing enzyme concentration and number of “holes” in the bioplastic as a result of enzymatic-mediated degradation (Figure 5). A non-enzyme/buffer-treated control of bioplastic was imaged as a ‘non-treated’ reference (Appendix A). ImageJ was used to quantify the number of holes produced between the enzyme and no-enzyme treatments [27]. As observed in Figure 5A–C, an increasing concentration of proteinase K resulted in an increase in the number of holes in the bioplastic. Specifically, in 3489 μm^2^ of the bioplastic surface, we observed the following: 47 holes with the no-enzyme control, 218 holes with 250 μg of proteinase K and 388 holes with 4000 μg of proteinase K (Figure 6A). A similar trend was observed when bioplastic was incubated with PLA depolymerase (Figure 5D–F): 22 holes were quantified in the no-enzyme control, 158 holes with 100 μg of PLA depolymerase, and 718 holes with 1000 μg of PLA depolymerase (Figure 6B). The visual observation of the “holes” created in the no-enzyme controls of this experiment provides further evidence of basal levels of bioplastic degradation during experiments in this study [22,29,30,31]. Although the entire 3489 μm^2^ of bioplastic was examined, there does appear to be local regions of high and low degradation.

Nuclear magnetic resonance (NMR) spectroscopy of the supernatants of in vitro reactions was used to observe bioplastic by-products produced through Proteinase K- and PLA depolymerase-mediated degradation (Appendix A). Specifically, we sought to determine if there was an increase in bioplastic by-products as a result of degradation, as well as identify the by-products. As PLA depolymerase acts as a hydrolase and sequentially hydrolyzes ester bonds down ester-containing bioplastic polymers, and a proteinase K-intrinsic mechanism of protein degradation is via Ser/Thr targeted degradation, we hypothesized that we would observe different end products through incubation with each respective enzyme [17,18]. As the authors were not aware of what the initial material of the commercially available bioplastic was, we were unable to identify the species of bioplastic by-products produced via enzymatic-mediated degradation. As for bioplastics incubated with PLA depolymerase, we observed an increase in three signals, which likely originate from degradation products (Appendix A). We were not able to observe similar by-products from bioplastic incubated with proteinase K (Appendix A). These results complemented that of Figure 3B where we saw a large increase in the A_610_ values in samples with a high concentration of PLA depolymerase. Although these results alone are not indicative of any signs of degradation, the authors feel the NMR results of the PLA depolymerase incubation in conjunction with the spectrophotometric-based assay, mass loss measurements and SEM data all show evidence that these enzymes degrade commercially available bioplastics.

It is well known that both temperature and co-factors such as Ca^2+^ can influence proteinase K activity [19,20,21]. In light of this, we sought to determine whether we could quantify proteinase K relative activity towards commercially available bioplastic using our spectrophotometric-based assay at different temperatures, as well as in the presence of 1 mM CaCl_2_ and 1 mM CaCl_2_ with 2 mM EDTA as a Ca^2+^ chelating agent [19,20,21]. Using the spectrophotometric-based assay, we observed how incubating proteinase K with commercially available bioplastic at temperatures of 37 °C and 65 °C affected the degradation of the bioplastics. Upon analyzing the residual bioplastic after the incubation of the bioplastics at varying proteinase K concentrations, there was an observed decrease in the relative A_610_ as proteinase K concentrations increased compared to the no-enzyme controls (Figure 7A). There were statistically significant differences in 37 °C vs. 65 °C temperatures in samples containing 62.5 μg/mL, 1000 μg/mL and 4000 μg/mL proteinase K but none at the other concentrations (*p* < 0.05). In analyzing the decanted supernatant upon reaction completion, we observed an increase in the relative A_610_ as proteinase K concentrations increased compared to the no-enzyme controls (Figure 7B). Consistent with the previous residual bioplastic analysis, we observed statistically significant differences in proteinase K activity between the relative A_610_ as a result of the difference in 37 °C vs. 65 °C incubation temperatures at higher concentrations of 1000 μg/mL and 4000 μg/mL proteinase K (*p* < 0.05). Although at both 37 °C and 65 °C the end-point activity curves followed the same trend of increased proteinase K activity as concentration increased, there were statistically significant differences in relative A_610_ values observed between many of the proteinase K concentrations data observed in both the residual bioplastic and supernatant sample. As such, the incubation temperature of 37 °C appears to be superior for the proteinase K-mediated degradation of commercially available bioplastic relative to higher temperatures of 65 °C. This is significant as it demonstrates that the spectrophotometric-based assay can be used to optimize bioplastic degradation over various temperatures.

Similar to the above, the spectrophotometric-based assay was used to observe the effects of incubating proteinase K with commercially available bioplastic on its own (enzyme only) in the presence of 1 mM CaCl_2_ (enzyme + CaCl_2_) and in the presence of 1 mM CaCl_2_ and 2 mM EDTA (enzyme + CaCl_2_ + EDTA) (Figure 7C,D). A two-way ANOVA was performed to compare the effect of how the three different reaction conditions alter proteinase K activity towards the bioplastic. For the residual bioplastic samples, generally, as Ca^2+^ concentration increased, there was a decrease in absorbance between our different enzyme reactions (Figure 7C). It appears that there was an interaction between the two independent variables between the concentration and the enzyme reaction condition; however, this was not statistically significant (F(2,66) = [3.12], *p* = 0.051). The addition of CaCl_2_ significantly altered the absorbance compared to the enzyme-only control (TukeyHSD, *p* = 0.002). The addition of CaCl_2_ and EDTA had no significant effect compared to the reactions just with CaCl_2_ (TukeyHSD, *p* = 0.126) nor compared to the enzyme-only control (TukeyHSD, *p* = 0.234). In analyzing the decanted supernatant upon reaction completion in comparison to the no-enzyme controls, we observed across all reaction conditions that there was an increase in the relative A_610_ as proteinase K concentrations increased (Figure 7D). Similar to the above, a two-way ANOVA was performed to compare the results between the reaction conditions, and there was not a significant interaction between our reaction conditions and increasing values of absorbance (F(2,66) = [2.47], *p* = 0.092). The addition of CaCl_2_ significantly altered the absorbance compared to the enzyme-only control (TukeyHSD, *p* = 0.002) and the CaCl_2_ and EDTA (TukeyHSD, *p* = 0.009). The addition of CaCl_2_ and EDTA had no significant effect compared to the enzyme-only control (TukeyHSD, *p* = 0.844). Overall, it appears that the addition of CaCl_2_ to the reactions did improve the activity of proteinase K towards the commercially available bioplastic in comparison to the enzyme-only control. This is significant as it demonstrates that the spectrophotometric-based assay can be used to evaluate how co-factors influence enzyme-mediated degradation of bioplastics.

## 4. Discussion

While there have been advancements in identifying enzymes that can degrade petroleum-based plastics and biodegradable plastics, limitations surrounding a medium-/high-throughput assay that can be used to screen candidate enzymes persist. Mass-loss measurements are commonly used, but these assays represent relatively low-throughput processes, require long assay times and relatively large reagent quantities, and require further assays or analytical methods to confirm the mass-loss data [12,14,22]. Other analytical approaches such as NMR, SEM, gel permeation chromatography and plate emulsification assays have assisted in finding candidate enzymes but may be cost-prohibitive and/or require long incubation/sample analysis time and specialized equipment [8,9,12,14,22]. Here, we demonstrate a novel, medium-throughput screening method that combines low volume and low sample requirements. The method has the advantage of assaying both residual remaining bioplastic and bioplastic by-products in order to screen candidate enzymes for bioplastic degradation. Using this method, we demonstrate the ability of candidate enzymes to degrade commercially available biodegradable plastic/bioplastics.

To date, a majority of research into plastic/bioplastic-degrading enzymes has been performed on pure plastic/bioplastic species. However, commercially available compostable bioplastics that comprise several plastic species are common pollutants in environments. Here, we show that the bioplastic-degrading enzymes proteinase K and PLA depolymerase degrade commercially available bioplastic after overnight incubation. We demonstrate statistically significant bioplastic degradation based on both reductions in residual bioplastic and the release of bioplastic degradation by-products into the supernatant in reactions with proteinase K concentrations >250 μg/mL (Figure 3A,B). Similarly, PLA depolymerase treatments resulted in a reduction in residual plastic and the release of degradation by-products at concentrations >100 μg/mL (Figure 3A,B). The degradation of commercial bioplastics by both enzymes was further confirmed by mass-loss and SEM assessment.

It should be noted that for proteinase K samples, there is a smaller increase in the relative A_610_ values for supernatant samples for both proteinase K concentration (Figure 3B) and reaction time (Figure 4B) experiments in comparison to the PLA depolymerase samples. This is most likely due to moieties in the commercial proteinase K enzyme preparation that lower the overall turbidity of the solutions upon emulsification, such as salts [32]. When both the 4000 μg/mL and 0 μg/mL proteinase K incubation time points were subtracted from the 0 μg/mL 0 h proteinase K reaction, we observe that the 4000 μg/mL proteinase K sample had a significantly lower raw A_610_ value compared to its no-enzyme control counterpart (Appendix A). This observation suggests that when any proteinase K is added to a solution that is emulsified with the emulsifying agents (CHCl_3_ and Triton X-100) used in this study, the solution becomes less turbid. As a result, this likely explains why the A_610_-based supernatant analysis yields less difference between higher and lower/no-enzyme concentrations for proteinase K than it does for PLA depolymerase and why those same effects are not observed in the residual bioplastic samples. This is significant as enzyme additives, whether commercial or added in-house, must be considered when applying this enzyme for screening purposes to ensure the collection of accurate data.

Supporting the literature suggesting the hydrolysis-susceptible nature of a variety of bioplastic species as a result of their ester-bond linkages, we observed low levels of the commercially available bioplastic degradation in all of our no enzyme-controlled experiments, hence the need for us to report all assay data in relative amounts [22,29,30,31]. The direct observation of background levels of bioplastic degradation can be made in both Figure 5A for the proteinase K and Figure 5D for the PLA depolymerase no-enzyme controls. It appears the added components (10% glycerol, 150 mM NaCl, 0.5 mM DTT) in the protein storage buffer in the PLA depolymerase no-enzyme control resulted in reduced basal levels of degradation compared to the 50 mM Tris-base buffer that was present in the proteinase K control. This could be attributed to the presence of NaCl in the protein storage buffer as it has been reported that some bioplastic species such as PLA do not degrade naturally in seawater [33,34,35]. Especially, the presence of salts in the seawater affects the diffusion of water into various biopolymers resulting in less natural hydrolysis and slower degradation [33,34].

We observed increases in PLA–depolymerase degradation by-products through the NMR analysis but not with proteinase K (Appendix A). A possible explanation is that proteinase K bioplastic degradation products may be much larger than those of PLA depolymerase. This may indicate that proteinase K cuts the bioplastic polymers internally rather than processively from ends. In this case, proteinase K by-products may be retained in the spin column used to remove remaining proteinase K from the supernatant upon the completion of the overnight incubation. The combination of our spectrophotometric-based assay and NMR may, therefore, provide interesting insights into the enzymatic degradation mechanisms of commercial bioplastics, although more research still needs to be conducted to confirm this theory.

## 5. Conclusions

Almost all studies to date have focused on the ability of enzymes to degrade pure bioplastic or petroleum-based single-use plastics [7,8,9,10,11,12,13,14,17,18]. Although this is critical information as it provides information on which enzymes and families of enzymes can target bioplastics, the practical application of these enzymes for commonly used bioplastics in society has not yet been fully investigated. Given the persistent problem of plastic pollution in a variety of waste systems, ecosystems and landscapes, it is critical to investigate enzymes that can degrade common commercially available bioplastics in a facile manner with minimal additives [2,7,22]. Moreover, it is critical to identify which enzymes can produce significant degradation in a short time frame (<24–48 h). We have demonstrated here the effectiveness of our spectrophotometric-based medium-throughput assay. This assay is ideal for screening candidate bioplastic degrading enzymes in a 24 h period. We have also demonstrated that it can be used to optimize specific reaction conditions such as temperature and the addition of co-factors. Although it is difficult to decipher what constitutes significant enzyme-mediated bioplastic degradation, our initial findings demonstrate a 20–30% mass reduction after the overnight incubation of proteinase K and PLA depolymerase. As a result of our findings, more research needs to be committed to optimizing the expression and yield of candidate enzymes, as well as to determine which would work optimally in an industrial-bioplastic-degrading system.

## Figures and Tables

**Figure 1 polymers-15-02439-f001:**
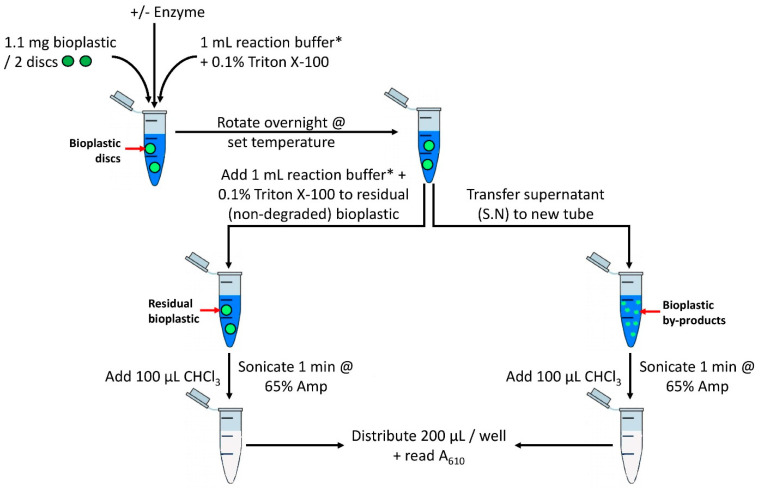
Spectrophotometric-based assay to quantify relative bioplastic degradation. Reaction setup and product detection for candidate bioplastic-degrading enzymes. Bioplastic degradation can be monitored by both reduction in residual plastic (bottom left) and increase in degradation products in supernatant (bottom right) by reading absorbance at 610 nm. Reaction includes 1.1 mg of plastic in 1 mL of reaction buffer with 0.1% Triton X-100 and absence or presence of enzyme of interest. In total, 10% *v*/*v* of CHCl_3_ is added as detection reagent. Solution is sonicated until uniformly emulsified. * Note: Reaction buffer will differ based on enzyme of interest.

**Figure 2 polymers-15-02439-f002:**
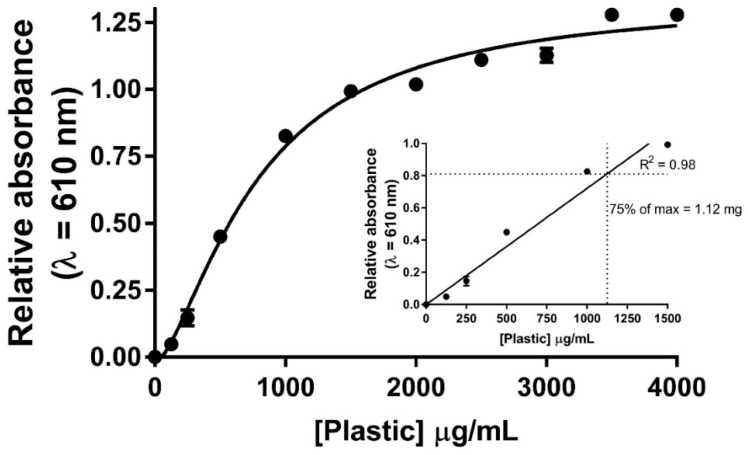
Standard curve for quantification of commercially available bioplastic. A_610_-based detection of different masses of plastic in 1 mL Tris-based buffer (50 mM Tris (pH 8.0) + 0.1% Triton X-100). CHCl_3_ (100 μL) was added as a detection reagent before emulsification via sonication of solution. Absorbance averages were normalized to no-plastic control signal. Results are mean ± std. error (n = 4). Plastic concentration vs. relative absorbance is fit with a nonlinear regression using a variable slope, with a least squares fit.

**Figure 3 polymers-15-02439-f003:**
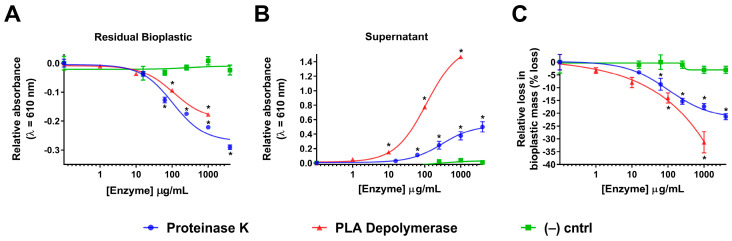
Proteinase K and PLA depolymerase activity towards commercially available bioplastic. (**A**). A_610_-based detection of residual commercially available bioplastic upon proteinase K, PLA depolymerase and negative control (lysing enzymes from *T. harzianum*) incubation with bioplastic. Absorbance averages were normalized by no-enzyme control signal. Results are mean ± std. error (n = 8). (**B**). A_610_-based detection of bioplastic by-product containing supernatant upon proteinase K, PLA depolymerase and negative control (lysing enzymes from *T. harzianum*) incubation with bioplastic. Absorbance averages were normalized by no-enzyme control signal. Results are mean ± std. error (n = 8). (**C**). Mass-loss of commercially available bioplastic upon proteinase K, PLA depolymerase and negative control (lysing enzymes from *T. harzianum*) incubation at 37 °C and 24 h. Mass-loss averages were normalized by no enzyme control signal. Results are mean ± std. error (n = 3). Enzyme concentration vs. relative absorbance data is fit with a nonlinear regression using a variable slope, with a least squares fit. * indicates significant difference from no-enzyme control (*p* < 0.05).

**Figure 4 polymers-15-02439-f004:**
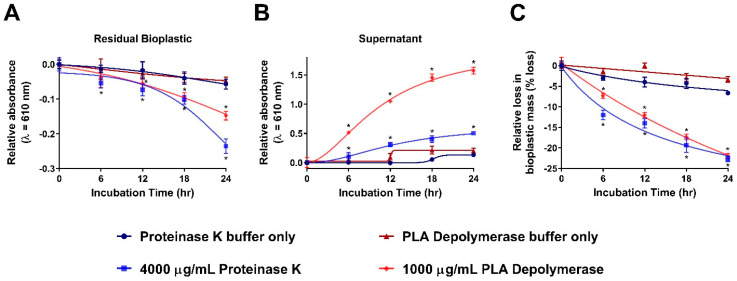
Temporal analysis of proteinase K and PLA depolymerase activity towards commercially available bioplastic. (**A**). A_610_-based detection of residual commercially available bioplastic upon reactions with proteinase K, PLA depolymerase and respective buffer only controls over 0, 6, 12, 18 and 24 h incubation periods. Enzyme-present absorbance averages were normalized by respective 0 h incubation control. Buffer-only absorbance averages were normalized by respective 0 h incubation control. Results are mean ± std. error (n = 8). (**B**). A_610_-based detection of bioplastic by-product containing supernatant upon reactions with proteinase K, PLA depolymerase and respective buffer only controls over 0, 6, 12, 18 and 24 h incubation periods. Enzyme-present absorbance averages were normalized by respective 0 h incubation control. Buffer-only absorbance averages were normalized by respective 0 h incubation control. Results are mean ± std. error (n = 8). (**C**). Mass-loss-based detection of residual commercially available bioplastic upon reactions with proteinase K, PLA depolymerase and buffer only controls over 0, 6, 12, 18 and 24 h incubation periods. Enzyme-present mass-loss averages were normalized by respective 0 h incubation control. Buffer-only absorbance averages were normalized by respective 0 h incubation control. Results are mean ± std. error (n = 3). Incubation time vs. relative absorbance data are fit with a nonlinear regression using a variable slope, with a least squares fit. * indicates significant difference from no-enzyme controls (*p* < 0.05).

**Figure 5 polymers-15-02439-f005:**
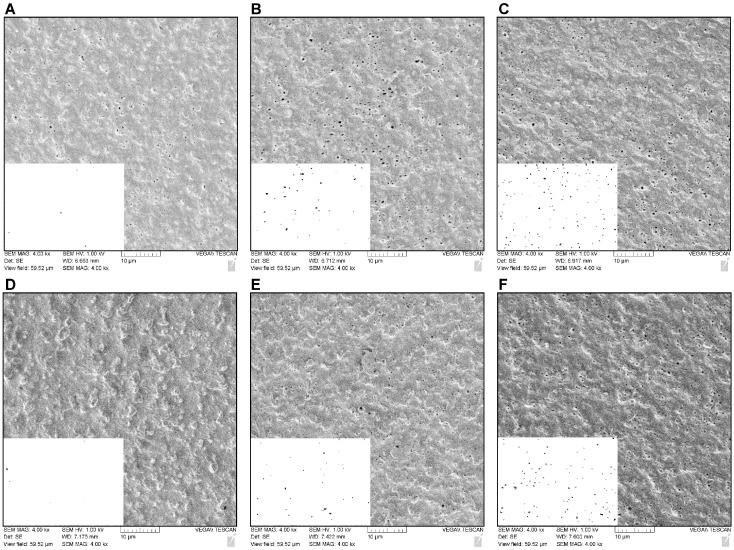
Scanning electron microscopy (S.E.M.) images of degraded commercially available bioplastic. (**A**) Bioplastic after 24 h incubation with Tris-based buffer. (**B**) Bioplastic after 24 h incubation with 250 μg/mL proteinase K in Tris-based buffer. (**C**). Bioplastic after 24 h incubation with 4000 μg/mL proteinase K in Tris-based buffer. (**D**). Bioplastic after 24 h incubation with protein storage buffer plus Triton X-100. (**E**). Bioplastic after 24 h incubation with 100 μg/mL PLA depolymerase in protein storage buffer plus Triton X-100. (**F**). Bioplastic after 24 h incubation with 1000 μg/mL PLA depolymerase in protein storage buffer plus Triton X-100. Bottom-left quadrant represents ImageJ “Analysis of particles” feature used for quantification of “holes” in bioplastic [27].

**Figure 6 polymers-15-02439-f006:**
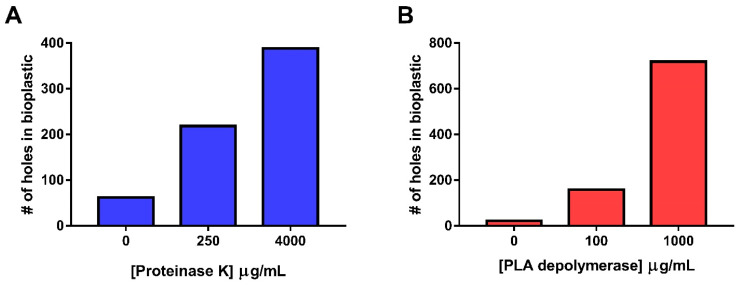
Quantification of number of “holes” detected by scanning electron microscopy (S.E.M.) per 3489 μm^2^ of commercially available bioplastic upon enzyme treatments. (**A**). Quantification of “holes” in bioplastic after 24 h incubation with and without proteinase K. (**B**). Quantification of “holes” in bioplastic after 24 h incubation with and without PLA depolymerase. Quantification analysis were conducted with S.E.M. images using Image J’s “Analysis of particles” feature [27].

**Figure 7 polymers-15-02439-f007:**
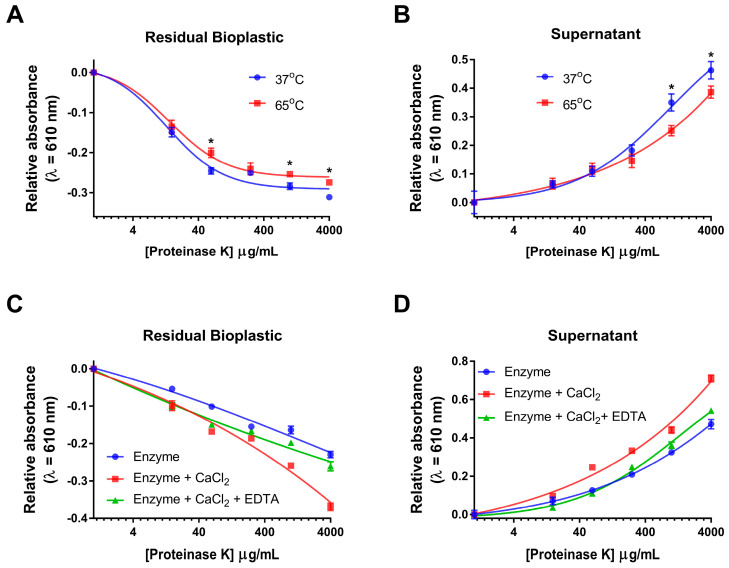
Effects of temperature and CaCl_2_ of proteinase K activity towards commercially available bioplastic. (**A**) A_610_-based detection of remaining commercially available bioplastic upon proteinase K incubation with bioplastic at 37 °C and 65 °C. Absorbance averages were normalized by no-enzyme control signal. Results are mean ± std. error (n = 4). (**B**) A_610_-based detection of bioplastic by-product containing supernatant upon proteinase K incubation with bioplastic at 37 °C and 65 °C. Absorbance averages were normalized by no-enzyme control signal. Results are mean ± std. error (n = 4). (**C**) A_610_-based detection of remaining commercially available bioplastic upon proteinase K incubation with no co-factors, 1 mM CaCl_2_ and 1 mM CaCl_2_ + 2 mM EDTA added. Absorbance averages were normalized by no-enzyme control signal. Results are mean ± std. error (n = 4). (**D**) A_610_-based detection of bioplastic by-product containing supernatant upon proteinase K incubation with no co-factors, 1 mM CaCl_2_ and 1 mM CaCl_2_ + 2 mM EDTA added. Absorbance averages were normalized by no-enzyme control signal. Results are mean ± std. error (n = 4). Enzyme concentration vs. relative absorbance data are fit with a nonlinear regression using a variable slope, with a least squares fit. * indicates significant differences at specified time points (*p* < 0.05).

## Data Availability

The data is not publicly available due to being already encapsulated in the figures of this manuscript. The data presented in this study are available on request from the corresponding author.

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
