# Peer review of "Spectrophotometric-Based Assay to Quantify Relative Enzyme-Mediated Degradation of Commercially Available Bioplastics"

_polymers, 2023, doi:10.3390/polym15112439_

Round 1

Reviewer 1 Report

Biodegradable bioplastics have begun to be introduced into various industries worldwide to replace petroleum-based plastics (PBPs). I think the research in this paper is very important and relevant for the earth's environment, but some modifications are needed before it can be accepted.

1. The authors in the abstract section present a lot of background of the study and lack a description of hydrolytic enzymes. The abstract section should also briefly introduce the experiments and experimental results so as to reflect the importance of the authors' research and increase the readability of the article. The determination based on spectrophotometry also needs to be explained in the abstract, which is the focus of the authors' study.

2. The introduction section lacks relevant literature on hydrolytic enzymes and spectrophotometric determinations, and the authors' research is mainly focused on the relevant sections.

3. in the material preparation section the authors could have made the preparation method into a table, which would have been better readable. The authors should add the relevant temperature at each step.

4. chapter 2 should be divided into sections, the material preparation section and the experimental section should be divided according to 2.1, 2.2, 2.3, etc.

5. In Figure 2, the authors mention that "once we determined the optimal amount of bioplastic for the assay enzyme (Figure 2), we evaluated the effectiveness of both enzymes". Was this determined for this study?

6. There does not seem to be much difference in the SEM images in Figure 5, it is suggested that the authors add distinctive markers in each figure to highlight the difference in the images.

7. The authors of the conclusion section can be streamlined and just highlight the experimental results and the significance of the experiment.

Language needs further optimization.

Reviewer 2 Report

 Fig 1, why in decanted solution lot of discs are shown. Mentioned Rotate solution in the figure is it correct.

Provide absorption plot to confirm the absorption maxima.

Holes in the SEM results what information?

Whether proteinase is stable at 67oC.

Suddenly in conclusion PLA-PET. What it is?

Satisfied

Reviewer 3 Report

I enjoyed reading this work, I liked the fact of developing a spectrophotometric-based medium-throughput assay to aid in screening enzymes that may degrade bioplastics. Also, thoroughly done and explained experiments are a very nice refresher in today's writing age.

Author Response

Thank you very much for the kind comments on the manuscript. 

Reviewer 4 Report

A well written manuscript, highlighting the need of robust screening method for the enzymatic degradation of biopolyester materials under different degradation conditions. The proposed medium throughput method for the determination of enzymatic degradation activity at the solid liquid interface is well described and of potential interest to people in the field (supported by the well written introduction). 

- Improvement might be possible in the graphical representations of the data (Fig. 2, Fig. 3( especially C) and Fig 4 (especially B)). It is not clear to the reviewer what the lines between the measured data are based on. It seems to be regressions without the explanation of the underlying mathematical functions.

- In addition The captions of the figures in the manuscript seems to be longer than necessary. Some of the repeated information on data treatment might be moved to the dedicated section of data treatment.   

minor spelling / grammar issues to be checked.

Round 2

Reviewer 2 Report

Authors modified the Ms properly based on reviewers comments.

Hence accepted for publication.